# Beyond Inhibition: A Novel Strategy of Targeting HIV-1 Protease to Eliminate Viral Reservoirs

**DOI:** 10.3390/v14061179

**Published:** 2022-05-28

**Authors:** Josh G. Kim, Liang Shan

**Affiliations:** Division of Infectious Diseases, Department of Medicine, Washington University School of Medicine, Saint Louis, MO 63110, USA; josh.kim@wustl.edu

**Keywords:** HIV/AIDS, protease, latent reservoir, CARD8 inflammasome, pyroptosis

## Abstract

HIV-1 protease (PR) is a viral enzyme that cleaves the Gag and Gag-Pol polyprotein precursors to convert them into their functional forms, a process which is essential to generate infectious viral particles. Due to its broad substrate specificity, HIV-1 PR can also cleave certain host cell proteins. Several studies have identified host cell substrates of HIV-1 PR and described the potential impact of their cleavage on HIV-1-infected cells. Of particular interest is the interaction between PR and the caspase recruitment domain-containing protein 8 (CARD8) inflammasome. A recent study demonstrated that CARD8 can sense HIV-1 PR activity and induce cell death. While PR typically has low levels of intracellular activity prior to viral budding, premature PR activation can be achieved using certain non-nucleoside reverse transcriptase inhibitors (NNRTIs), resulting in CARD8 cleavage and downstream pyroptosis. Used together with latency reversal agents, the induction of premature PR activation to trigger CARD8-mediated cell killing may help eliminate latent reservoirs in people living with HIV. This represents a novel strategy of utilizing PR as an antiviral target through premature activation rather than inhibition. In this review, we discuss the viral and host substrates of HIV-1 protease and highlight potential applications and advantages of targeting CARD8 sensing of HIV-1 PR.

## 1. Introduction

The acquired immunodeficiency syndrome (AIDS) global pandemic is one of the most devastating public health challenges in human history, with more than 36 million people estimated to have died from AIDS-related illnesses since the beginning of the epidemic in 1981 [1]. In the early 1980s, the novel retrovirus human immunodeficiency virus type 1 (HIV-1) was identified as the causative agent of AIDS [2,3,4]. The HIV-1 genome encodes capsid proteins (Gag), viral enzymes (Pol), and the envelope glycoprotein (Env), as well as six regulatory proteins. The viral enzymes encoded by *pol* are protease (PR), reverse transcriptase (RT), and integrase (IN), which were targets of some of the first antiviral inhibitors for treating HIV [5]. The first antiretroviral drug to be approved by the US Food and Drug Administration (FDA) as a treatment for HIV was zidovudine (AZT, a nucleoside reverse transcriptase inhibitor (NRTI)), in 1987. This was followed by the approval of several other NRTIs such as didanosine, zalcitabine, and stavudine. The FDA approval of the first protease inhibitor (PI) saquinavir (SQV) in 1995 marked a watershed moment in the development of antiretroviral therapy (ART). ART is a medication regiment consisting of a combination of several antiretroviral drugs targeting different stages of the HIV life cycle [6]. With the advent of ART, people living with HIV (PLWH) can live longer and healthier lives [7]. While ART can effectively control viral replication in PLWH, it cannot eradicate the virus due to the latent reservoir, a pool of latently infected, resting CD4^+^ T cells. Upon discontinuation of ART, the latent reservoir can quickly re-establish viral replication [8,9,10,11]. Therefore, PLWH must take ART daily for life. According to the World Health Organization, there were about 38 million people globally living with HIV in 2021 [1].

During viral maturation, PR sequentially cleaves the Gag and Gag-Pol polyproteins at nine cleavage sites to convert them into their functional forms [12]. As this step is required to generate infectious viral particles, PR is an essential viral enzyme and thus an attractive target for anti-HIV drug design [13]. The currently approved HIV drugs that target PR are all compounds that mimic the substrate transition site of HIV PR to inhibit its activity [14]. Given the emergence of drug-resistant strains of HIV-1, novel strategies of targeting HIV PR are urgently needed. Interestingly, HIV PR has been shown to interact with numerous host cell proteins [15,16,17,18]. As the cleavage of certain host substrates by HIV-1 PR has been linked to cell death, targeting the interactions between PR and the host rather than PR and viral polyprotein precursors may open promising new avenues for antiviral therapy. For instance, recent evidence indicates HIV-1-infected cells may be cleared by the targeted activation of PR, which cleaves and activates the caspase recruitment domain-containing protein 8 (CARD8) inflammasome to induce cell death [19]. In this review, we discuss the virus- and host-derived substrates of HIV-1 PR and highlight novel strategies of targeting PR for an HIV cure.

## 2. HIV Protease Structure, Function, and Inhibitors

Decades of structural biology and X-ray crystallography research have provided a thorough understanding of the structure and function of HIV PR. HIV PR exists as a homodimer consisting of two identical subunits of 99 amino acids. It is a member of the aspartic protease family, with a conserved catalytic Asp residue at position 25 [20]. The Asp25 and Asp25′ residues from each monomer meet at the dimer interface and form the enzyme active site; dimerization is required for PR activation as monomers of HIV PR are enzymatically inactive [21,22]. The active site is covered by two flexible glycine-dense β-sheet flaps. Upon substrate binding, these flaps undergo a conformational shift that causes the flaps to close and cover the active site [23].

In the HIV life cycle, the cleavage of several peptide bonds in viral polyprotein precursors by PR is necessary to produce mature active enzymes. Interestingly, HIV PR has broad specificity which is determined by the asymmetric shape of the substrate rather than a particular amino acid sequence. This allows PR to cleave Gag and Gag-Pol at multiple sites with different amino acid sequences [24]. However, differences in amino acid side chains at different cleavage sites may lead to small structural differences that could affect the rate of cleavage at individual sites and contribute to sequential processing of Gag and Gag-Pol [25,26]. Importantly, in HIV-1-infected cells, PR exists as a subunit of the Gag-Pol polyprotein precursor with minimal proteolytic activity and must dimerize to become catalytically active. The timing of PR activation is tightly regulated, as premature or delayed PR activation results in dramatically decreased infectivity [27,28,29,30]. According to a recent study on the kinetics of PR activation, HIV-1 PR becomes activated during viral assembly and budding, just prior to virion release [31].

The elucidation of PR structure through crystallography, NMR, and computational biochemistry methods led to a monumental effort of structure-based rational drug design for antivirals to treat HIV in the early to mid 1990s [32]. Most of these drug discovery approaches were based on the synthesis of peptide substrate analogs designed to mimic the substrate transition state. The contact points between HIV PR and most substrate analog inhibitors are quite similar, with a key interaction being the hydrogen bonds between carbonyl oxygens of PR active site residues Asp25 and Asp25′ and the hydroxyl groups of the inhibitor [20].

Early clinical trials showed remarkable reductions in morbidity, mortality, and HIV viral load in PLWH treated with a combination of a PI and NRTIs, demonstrating the effectiveness of combination therapy in HIV treatment [7,33,34]. There are currently six PIs approved and recommended for use by the FDA: atazanavir, darunavir, fosamprenavir, ritonavir, saquinavir, and tipranavir [14]. Some other PIs, such as lopinavir, are recommended as part of combination HIV medicines such as ritonavir-boosted lopinavir. Unfortunately, HIV PIs are frequently associated with side effects resulting from drug–drug interactions, overdose, or interactions with off-target molecules [20]. The emergence of PI drug resistance also poses a major problem in the long-term effectiveness of PIs as an anti-HIV drug [35]. One approach to combat drug resistance has been to boost with ritonavir, which increases the circulating concentration of other PIs by inhibiting cytochrome P450 3A4 [36]. Another major strategy has been to design PIs that incorporate chemical groups that promote hydrogen bonding with backbone atoms in the active site of HIV PR [22].

## 3. Host Cell Substrates of HIV-1 PR

In addition to cleaving viral polyprotein precursors, HIV PR cleaves a wide array of host cell proteins. Indeed, as many as 123 human proteins were identified as substrates of HIV PR in a 2012 study that incubated lysates of Jurkat T-cells with HIV PR and utilized mass spectrometry to identify substrate identity and the cleavage site location [15]. The cleavage of host proteins by HIV-1 PR may have some impact on the course of HIV infection. For instance, HIV-1 PR can cleave the cytoskeletal protein vimentin, resulting in the release of an N-terminal fragment that leads to changes in nuclear architecture [37,38]. These effects are likely harmful to the cell, although the exact role of vimentin cleavage in HIV infection is still unclear. Furthermore, eukaryotic translation initiation factor 3d (eIF3d) was found to be cleaved by HIV PR both in a HEK293 cell transfection system and in vitro using purified eIF3d incubated with PR [39]. eIF3 is a translational initiation factor involved in translation initiation, termination, and ribosomal recycling [40]. Knockdown of eIF3d—but not other subunits of eIF3—resulted in an increase in HIV-1 infectivity, suggesting the cleavage of eIF3d by HIV-1 PR may be a strategy for HIV to overcome inhibition of viral replication by eIF3 [39]. eIF4G, another eukaryotic translation initiation factor, was also found to be cleaved by HIV-1 PR in a study that tested eIF4G cleavage in HIV-1-infected CD4^+^ T cells and in vitro using purified eIF4G [16]. The cleavage of eIF4G by HIV-1 PR was shown to inhibit cap-dependent cellular translation, providing a potential mechanism by which HIV-1 may modulate host protein synthesis [16].

The cleavage of certain host proteins by HIV-1 PR has been linked to cell death. A 1996 study reported that the transfection of various cell lines with a plasmid coding for HIV-1 PR resulted in increased apoptosis which was associated with the cleavage of endogenous BCL-2 by PR [17]. BCL-2 family proteins are known to mediate cell death, with the overexpression of *BCL-2* linked to the inhibition of apoptosis [41]. Correspondingly, co-transfection of BCL-2 with HIV-1 PR in COS-7 cells protected against apoptosis and reduced HIV infectivity, p24 and gp120 levels, and tumor necrosis factor alpha (TNFα) release [17]. The authors speculate that BCL-2 may suppress the production of reactive oxygen species (ROS) and the cleavage of BCL-2 by HIV-1 PR could therefore lead to accumulation of ROS, activation of nuclear factor kappa B (NF-κB), and increased HIV-1 transcription. However, as this study only utilized an overexpression model in cell lines, its relevance to in vivo HIV-1 infection is unclear, especially considering the lower level of PR activity in naturally infected cells.

In a cell-free system using cytoplasmic extracts from Jurkat T cells, HIV-1 PR was shown to cleave and activate pro-caspase-8, an initiator caspase of extrinsic apoptosis [18]. The cleavage of pro-caspase-8 in cell extracts was associated with the cleavage of BID, a member of the BCL-2 family that is known to be cleaved by caspase-8 [42]. Pro-caspase-8 cleavage was also associated with mitochondrial cytochrome c release, consistent with previous data showing that truncated BID (tBID) translocates into the mitochondria and induces cytochrome c release into the cytosol [43]. Notably, the HIV-1 PR cleavage site of pro-caspase-8 differs from the typical cleavage site, as a mutation of the normal pro-caspase-8 cleavage site did not affect cleavage patterns by HIV-1 PR [18]. A follow-up study showed that transfection of the novel fragment produced by HIV-1 PR cleavage of pro-caspase-8 (termed casp8p41) into both HeLa cells and primary CD4^+^ T cells caused apoptosis [44]. Further, casp8p41 expression was detected in HIV-1-infected Jurkat T cells, but not mock-infected cells, and the proportion of cells expressing casp8p41 was positively correlated with the proportion of cells expressing p24 and with cell death. In vivo, casp8p41 was not detected in peripheral blood mononuclear cells from people without HIV but was detected in CD4^+^ T cells from PLWH [44]. While the cleavage of pro-caspase-8 by HIV-1 PR represents a potential mechanism by which HIV-1 induces apoptosis in infected cells, the importance of this mechanism in natural HIV-1 infection remains inconclusive.

Host cell protein cleavage by viral PR may represent a strategy for the virus to counteract host immune mechanisms. For instance, HIV RNA transcripts contain numerous post-translational modifications, including N6-methyladenosine (m^6^A). Reader proteins recognize these post-translational modifications and fulfill their functional outcomes. A recent study showed that YTHDF3, a reader protein for m^6^A, is incorporated into HIV particles [45]. YTHDF3-deficient cells were more susceptible to HIV infection, suggesting an antiviral function of YTHDF3. HIV-1 PR was found to cleave virion-incorporated YTHDF3, representing a potential mechanism by which HIV may counter the antiviral activity of YTHDF3. Interestingly, YTHDF3 has also been described to be targeted by enterovirus 2A proteases [46]. Another example is the receptor interacting protein kinase (RIPK) family, which play important roles in responses to a wide array of viral infections [47]. Several viruses have evolved strategies to disrupt RIPK function and evade RIP-dependent cell death [48,49,50]. In the context of HIV infection, one study demonstrated that infection of primary CD4^+^ T cells with replication-competent HIV-1 results in the cleavage of endogenous RIPK1 and RIPK2 by the viral PR [48]. This cleavage may disrupt the function of RIPK1, as a co-overexpression of RIPK1 and HIV-1 PR in HEK293T cells resulted in a diminished ability of RIPK1 to activate NF-κB; however, the disruption of RIPK1 function during viral infection was not tested [48]. The physiological role of RIPK1 and RIPK2 cleavage by HIV-1 PR remains unclear, as no RIPK family proteins have been linked to protection against HIV-1 infection [48].

## 4. HIV-1 PR and the CARD8 Inflammasome

Of particular interest is the recent discovery that the caspase recruitment domain-containing the protein 8 (CARD8) inflammasome can be cleaved and activated by HIV-1 PR [19]. Inflammasomes are multiprotein complexes that function as innate sensors of pathogenic microorganisms and tissue damage [51]. They are assembled by pattern-recognition receptors (PRRs), which become activated upon detection of pathogen-associated molecular patterns (PAMPs) or cytosolic danger signals. Activated receptors undergo oligomerization and recruit apoptosis-associated speck-like protein containing a CARD (ASC), an adaptor protein which consists of a pyrin domain (PYD) and a caspase recruitment domain (CARD). ASC helps bridge the sensor molecule to pro-caspase-1 (pro-CASP1), where proximity-induced autoprocessing generates the catalytically active caspase-1 (CASP1). Interestingly, some CARD-containing PRRs can directly recruit pro-CASP1 without the need for ASC [52,53,54]. Activation of CASP1 initiates pyroptosis, an inflammatory form of programmed cell death.

In the canonical pyroptotic cell death pathway, activated CASP1 processes the inactive precursors of the inflammatory cytokines interleukin-1β (IL-1β) and IL-18 into their mature forms [55]. CASP1 also cleaves the protein gasdermin D, which then forms 10–20 nm diameter pores in the plasma membrane [56,57]. These pores facilitate secretion of IL-1β and IL-18 and dissipate cellular ionic gradients, resulting in water influx, cell swelling, and osmotic lysis [58]. DNA damage occurs during pyroptosis as CASP1 stimulates an unidentified nuclease which causes DNA fragmentation [57]. Unlike in apoptosis, however, the nucleus remains intact in cells undergoing pyroptosis [59].

In 2002, a study reported a new CARD family member called CARD8 which could interact physically with CASP1, regulate CASP1 activation, and cause apoptosis when overexpressed [60]. A later cancer study showed that dipeptidyl peptidase 8/9 (DPP8/9) inhibitor-induced pro-CASP1-dependent pyroptosis is mediated by CARD8, implicating it for the first time as an inflammasome sensor, though its natural ligands, function, and mechanism of activation were still unclear [61]. CARD8 consists of a function-to-find (FIIND) domain followed by a CARD domain and is one of only two proteins in the human genome containing FIIND-CARD, the other being NACHT, LRR, and PYD domains-containing protein 1 (NLRP1). The FIINDs of CARD8 and NLRP1 consist of ZU5 and UPA subdomains; following translation, CARD8 and NLRP1 undergo intramolecular autoproteolysis after their ZU5 domains, resulting in two non-covalently associated polypeptide chains [62]. This autoprocessing is required for both NLRP1 and CARD8 inflammasome activation [19,63]. The human NLRP1 inflammasome can sense and be activated by enteroviral protease and double-stranded RNA, but its interaction with HIV-1 remains unclear [64,65]. It was recently shown that HIV-1 PR directly cleaves the N-terminus of CARD8 [19]. This cleavage causes the release of an unstable neo-N-terminus which is targeted for proteasomal degradation. If the CARD8 molecule has undergone FIIND autoprocessing, the break in the polypeptide chain causes the C-terminal UPA-CARD fragment to be liberated from the proteosome. The bioactive UPA-CARD fragment directly recruits and activates pro-CASP1, which leads to the secretion of inflammatory cytokines and ultimately pyroptotic cell death (Figure 1).

HIV-1 PR cleavage of CARD8 is physiologically relevant because CARD8 is expressed in cells targeted by HIV-1, including primary CD4^+^ T cells and macrophages [19]. However, as PR has low catalytic activity prior to virion release [31], HIV-1-infected cells can evade CARD8 sensing and cell death; thus, premature activation of PR is needed to induce CARD8-mediated CASP1 activation and pyroptosis of infected cells. Premature PR activation can be achieved either through PR overexpression or the use of certain non-nucleoside reverse transcriptase inhibitors (NNRTIs) which have been shown to bind to HIV-1 Pol and enhance Gag-Pol dimerization [66]. Indeed, the treatment of HIV-1-infected macrophages and primary CD4^+^ T cells with the NNRTIs efavirenz (EFV) or rilpivirine (RPV) results in profound and rapid pyroptosis of infected cells, which is abrogated in *CARD8*-KO cells [19]. Moreover, treatment with EFV and RPV was found to reduce the size of viral reservoirs in a quantitative viral outgrowth assay using blood CD4^+^ T cells from PLWH under ART [19]. Importantly, CARD8 was shown to sense all subtypes of HIV-1 despite significant viral diversity, demonstrating that CARD8 senses PR function rather than any specific sequence [19]. This offers a distinct advantage over other immune-based strategies of targeting HIV-1, such as antibody or T-cell treatment, which rely on recognition of highly variable viral epitopes [67,68,69]. Together, these data suggest targeted activation of the CARD8 inflammasome may be a promising strategy to eliminate HIV-1-infected cells. Combined with the use of latency reversal agents (LRAs) to reactivate HIV-1 viral gene expression in latently infected cells, HIV-1 PR activation and subsequent sensing and downstream cell death by CARD8 may help clear residual viral reservoirs in PLWH, the major obstacle to an HIV cure (Figure 1).

It should be noted that excessive pyroptosis has the potential to induce pathological inflammation that may fuel HIV disease progression or contribute to the development of other chronic diseases [55,70]. Pyroptosis of residual infected CD4^+^ T cells in PLWH is unlikely to result in substantial inflammation because CD4^+^ T cells generally do not produce IL-1β or IL-18 [71]. Moreover, the frequency of latently infected CD4^+^ T cells is very low, with approximately 1 in 10^5^ to 10^8^ latently infected cells in most PLWH [72]. However, recent evidence points to the potential for long-lived tissue macrophages to be an HIV reservoir [73]. Macrophages can reside in virtually every tissue and release IL-1β and IL-18 upon CASP1 activation. Therefore, inducing pyroptosis of latently infected macrophages could result in local inflammation depending on the size of macrophage reservoirs in PLWH. Further studies on macrophage reservoirs are needed to predict the safety and potential side effects of inducing pyroptosis through CARD8 activation.

## 5. Conclusions and Future Perspectives

HIV-1 PR is a valuable target for antiviral strategies because of its indispensable function in the HIV life cycle. The development of the first HIV PIs led to a revolution in HIV treatment, with ART allowing HIV to be a manageable health condition. However, ART must be taken daily for life due to the persistence of the latent reservoir which can quickly rekindle infection if ART is stopped. Recent evidence showing that HIV-1 PR can be sensed by the CARD8 inflammasome and trigger cell death opens new possibilities for targeting PR to eliminate the latent reservoir and move toward an HIV cure. For example, the awakening of latently infected cells using LRAs followed by treatment with an NNRTI to cause premature PR activation should lead to CARD8-mediated clearance of viral reservoirs.

Some obstacles toward implementation of this strategy include the high concentrations of NNRTIs required for CARD8 inflammasome activation in HIV-1-infected cells and the fact that NNRTIs bind human serum proteins, reducing their bioavailability in vivo [74]. A recent preprint suggests that CARD8 sensitization through inhibition of DPP9, a negative regulator of CARD8, can overcome these obstacles and enhance NNRTI-triggered killing of HIV-1-infected cells [75]. Furthermore, drug screening for compounds that more efficiently trigger Gag-Pol dimerization and premature PR activation is a promising strategy to identify novel antivirals that can efficiently induce CARD8 inflammasome-dependent killing of HIV-1-infected cells with few side effects. Finally, it is possible that the strategy to induce CARD8 activation and cell killing through premature PR activation could be applied to other viruses. As HIV-1 PR shares high sequence homology with PR from other retroviruses such as HIV-2 and Human T-lymphotropic virus (HTLV), future studies should investigate whether these viral proteases have the ability to cleave and activate the CARD8 inflammasome.

## Figures and Tables

**Figure 1 viruses-14-01179-f001:**
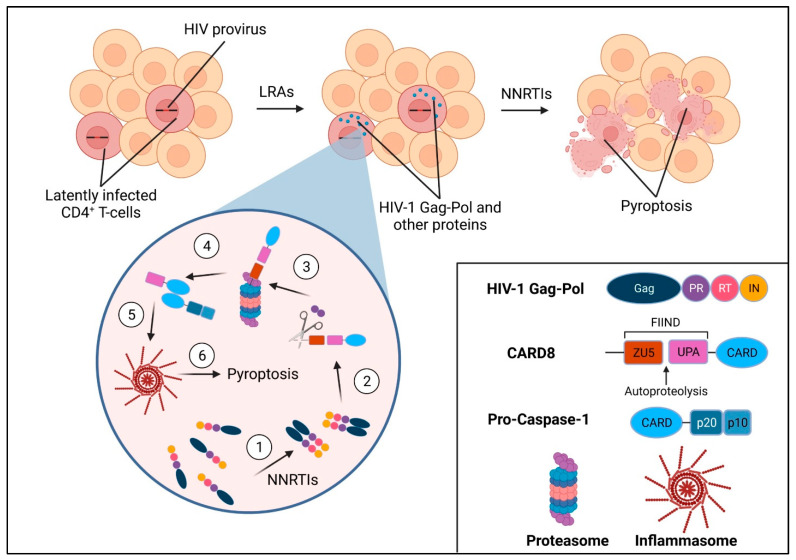
The shock and kill scheme using targeted CARD8 activation. Latency reversal agents (LRAs) are used to reactivate viral gene transcription in latently infected resting memory CD4^+^ T cells, leading to production of HIV Gag-Pol polyprotein and other viral proteins. The treatment with non-nucleoside reverse transcriptase inhibitors (NNRTIs) leads to Gag-Pol dimerization and premature PR activation (1). The activated PR cleaves the N-terminus of CARD8 (2), which causes the neo-N-terminus to be targeted for proteasomal degradation (3). Due to the break in the polypeptide chain between the ZU5 and UPA domains of CARD8 caused by autoproteolysis, the bioactive UPA-CARD fragment is released from the proteasome (4). The UPA-CARD fragment recruits and activates pro-caspase-1 (4), leading to downstream inflammasome assembly (5) and pyroptosis (6).

## Data Availability

Not applicable.

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
