# Peer review of "Beyond Inhibition: A Novel Strategy of Targeting HIV-1 Protease to Eliminate Viral Reservoirs"

_viruses, 2022, doi:10.3390/v14061179_

Round 1

Reviewer 1 Report

The review written by Kim and Shan is well written and brings up an interesting concept that HIV gene activation by drugs can be utilized in cell killing by HIV PR overactivation caused by drugs like HIV IN inhibitors causing enhanced Pol dimerization. Though this is an interesting concept, the potencial danger of the procedure should be emphasized.

Minor points:

Also, it is a misconception that the PR precursor does not have activity, that would prevent autoprocessing, although it is much smaller that that of the fully autoprocessed PR. (e.g. Louis et al,  Nat. Struct Biol, 1999).

It is not clear how eIF3f is relevant in relation of the PR activity? (page 3) and why eIF3G is given with capital G unlike all the others?

Author Response

The review written by Kim and Shan is well written and brings up an interesting concept that HIV gene activation by drugs can be utilized in cell killing by HIV PR overactivation caused by drugs like HIV IN inhibitors causing enhanced Pol dimerization. Though this is an interesting concept, the potential danger of the procedure should be emphasized.

We thank the reviewer for bringing up the important issue of potential side effects of inducing pyroptosis of HIV-infected cells through the CARD8 pathway. The side effects of inflammasome activation is due to caspase 1-dependent cleavage of por-IL-1β and pro-IL-18 and subsequent release of bioactive IL-1β and IL-18. However, CD4+ T cells generally do not produce IL-1β or IL-18. In addition, the frequency of residual infected CD4+ T cells in people living with HIV on ART is around 1 in 105-108 total CD4+ T cells. Therefore, CARD8-mediated pyroptosis of HIV-1-infected CD4+ T cells is unlikely to result in substantial inflammation. However, pyroptosis of tissue macrophage reservoirs of HIV is a potential concern because macrophages do produce por-IL-1β and pro-IL-18 and can release active forms of IL-1β and IL-18 during pyroptosis. These discussions have been added in lines 229-239 of the revised manuscript (highlighted in red). 

Minor points:

Also, it is a misconception that the PR precursor does not have activity, that would prevent autoprocessing, although it is much smaller that that of the fully autoprocessed PR. (e.g. Louis et al, Nat. Struct Biol, 1999).

We thank the reviewer for pointing out this misconception. We have re-worded the sentences in lines 75-80 (highlighted in red).

It is not clear how eIF3f is relevant in relation of the PR activity? (page 3) and why eIF3G is given with capital G unlike all the others?

We thank the reviewer for the comments. eIF3f is not related to HIV PR activity. We have removed this sentence. Regarding the capital G in eIF4G: eIF3 complex subunits, such as eIF3d, are usually given with lower case letters (PMID 33184215) whereas eIF4 complex subunits, such as eIF4G, are usually given with capital letters (PMID 33783376).

Reviewer 2 Report

This review from Kim and Shan describes the mechanism of action of some host factors (including CARD8) facilitating HIV infection upon cleavage by the HIV protease. These factors would help as tools or targets to reduce HIV reservoirs, for instance, in the shock and kill HIV eradication strategy. Although the review is interesting, some significant changes would improve the review:

1-Authors emphasized the CARD8 factor in their review; however, several beautiful reviews were recently published on CARD8, which decreases the enthusiasm for this review (Sparrer &Kirchhoff, Nature 2021, Jin et al., cell 2022, Tsu et al., Front. Immunol., 2021…). Therefore, this paper would benefit from adding a more comprehensive list of viral/host factors, including YTHDF3, RIPK1/K2, and NLRP1. Authors should discuss their combination and action on other viruses.

2-Authors should also discuss a chronic state of inflammation induced by pyroptosis that would fuel HIV disease progression, premature onset of other end-organ diseases, and cognitive impairments.

3-Authors should add a paragraph explaining the pyroptosis death pathway.

4-The English needs a substantial make-over with some terms that must be changed, such as patients should be replaced by "people/individuals living with HIV". There is also a lack of consistency, such as ART/HAART. Finally, some sentences are too long such as "while the use of HAART can effectively……upon discontinuation of HAART".

Author Response

This review from Kim and Shan describes the mechanism of action of some host factors (including CARD8) facilitating HIV infection upon cleavage by the HIV protease. These factors would help as tools or targets to reduce HIV reservoirs, for instance, in the shock and kill HIV eradication strategy. Although the review is interesting, some significant changes would improve the review.

We thank the reviewer for their constructive comments and are glad they found the review interesting.

1-Authors emphasized the CARD8 factor in their review; however, several beautiful reviews were recently published on CARD8, which decreases the enthusiasm for this review (Sparrer &Kirchhoff, Nature 2021, Jin et al., cell 2022, Tsu et al., Front. Immunol., 2021…). Therefore, this paper would benefit from adding a more comprehensive list of viral/host factors, including YTHDF3, RIPK1/K2, and NLRP1. Authors should discuss their combination and action on other viruses.

Requested by the Special Issue “Enzymes as Antiviral Targets”, our review article focuses on HIV PR activity and its host cell substrates, which is different from other review articles mentioned by the Reviewer. We agree with the Reviewer that adding a more comprehensive list of host cell substrates of HIV PR would improve the review. We have included discussion about HIV PR cleavage of vimentin, YTHDF3, RIPK1, and RIPK2 in lines 107-110 and 150-167 (highlighted in red). To date, HIV PR is the only physiological ligand for CARD8. Thus, we are unable to discuss the interaction between CARD8 and other viruses.

By the way, we cannot find any report on the interaction between HIV PR and NLRP1. We added some discussion about NLRP1 in lines 200-201 (highlighted in red).  

2-Authors should also discuss a chronic state of inflammation induced by pyroptosis that would fuel HIV disease progression, premature onset of other end-organ diseases, and cognitive impairments.

We thank the reviewer for highlighting this point and agree that it is important to acknowledge the potential side effects of inducing CARD8 activation and pyroptosis. The side effects of inflammasome activation is due to caspase 1-dependent cleavage of por-IL-1β and pro-IL-18 and subsequent release of bioactive IL-1β and IL-18. However, CD4+ T cells generally do not produce IL-1β or IL-18. In addition, the frequency of residual infected CD4+ T cells in people living with HIV on ART is around 1 in 105-108 total CD4+ T cells. Therefore, CARD8-mediated pyroptosis of HIV-1-infected CD4+ T cells is unlikely to result in substantial inflammation. However, pyroptosis of tissue macrophage reservoirs of HIV is a potential concern because macrophages do produce por-IL-1β and pro-IL-18 and can release active forms of IL-1β and IL-18 during pyroptosis. These discussions have been added in lines 229-239 of the revised manuscript (highlighted in red). 

3-Authors should add a paragraph explaining the pyroptosis death pathway.

We thank the reviewer for this suggestion. A paragraph explaining the pyroptotic death pathway has been added in lines 181-188 (highlighted in red).

4-The English needs a substantial make-over with some terms that must be changed, such as patients should be replaced by "people/individuals living with HIV". There is also a lack of consistency, such as ART/HAART. Finally, some sentences are too long such as "while the use of HAART can effectively……upon discontinuation of HAART".

We thank the reviewer for their attention to these issues and hope the reviewer will be satisfied with the changes that have been made to improve the consistency and wording of the manuscript.

Round 2

Reviewer 2 Report

The authors have satisfactorily responded to our comments and made the appropriate edits to the review.